# Changes of Estimated Glomerular Filtration Rate and Glycated Hemoglobin A1c in Diabetic Macular Edema Patients Treated by Ranibizumab and Aflibercept in the Tertiary Referral Hospital

**DOI:** 10.3390/medicina58081081

**Published:** 2022-08-11

**Authors:** Wei-Ning Ku, Peng-Tai Tien, Chun-Ju Lin, Chun-Chi Chiang, Ning-Yi Hsia, Chun-Ting Lai, Chih-Hsin Muo, Henry Bair, Huan-Sheng Chen, Jane-Ming Lin, Wen-Lu Chen, Yi-Yu Tsai

**Affiliations:** 1Department of Ophthalmology, Eye Center, China Medical University Hospital, China Medical University, 2 Yuh-Der Road, Taichung 40477, Taiwan; 2School of Medicine, College of Medicine, China Medical University, Taichung 40042, Taiwan; 3Department of Optometry, Asia University, Taichung 41354, Taiwan; 4Management Office for Health Data, China Medical University Hospital, Taichung 40459, Taiwan; 5Byers Eye Institute, Stanford University School of Medicine, Stanford, CA 94303, USA; 6An-Shin Dialysis Center, NephroCare Ltd., Fresenius Medical Care, Taichung 43655, Taiwan; 7School of Chinese Medicine, College of Chinese Medicine, China Medical University, Taichung 40402, Taiwan

**Keywords:** HbA1c, DME, anti-VEGF, IVI, eGFR, intravitreal injections, ranibizumab, aflibercept

## Abstract

*Background and Objectives*: Intravitreal injections (IVI) of vascular endothelial growth factor (VEGF) inhibitors are guideline-indicated treatments for diabetic macular edema (DME). However, some recent data have suggested that IVI VEGF inhibitors might, through systemic absorption, lead to a reduction in renal function. Our study aims to compare changes in glycated hemoglobin A1c (HbA1c) and estimated glomerular filtration rate (eGFR) between patients who received IVI ranibizumab and aflibercept treatment and patients who have not received IVI treatments. *Materials and Methods*: There were 17,165 DME patients with documented ophthalmology visits in the China Medical University Hospital-Clinical Research Data Repository. Those with a history of ESRD or bevacizumab treatment history, and those with missing information on HbA1c or eGFR, were excluded. After matching by age (±2 years), gender, and the year of clinical visit, 154 patients with medical treatment (including ranibizumab and aflibercept) and 154 patients without medical treatment were included in the study. The difference between HbA1c and eGFR at baseline and 3 and 12 months after the index date between the two groups was assessed. *Results*: Mean HbA1c and eGFR decreased between baseline and 12 months after the index date in both groups (*p* < 0.05). Compared with the non-treatment group, the treatment group had significantly lower HbA1c 3 and 12 months after the index date. There was no significant difference in eGFR between the two groups. In the generalized estimating equations (GEE) model, HbA1c in the treatment group was lower than the non-treatment group (−0.44%, 95% CI = −0.75, −0.14), but eGFR was similar after adjusting for age, gender, and index-year. HbA1c and eGFR decreased with the time in the adjusted GEE model (*p* < 0.0001) in both groups. *Conclusions*: This study showed that eGFR decreased with age and time and was not related to IVI anti-VEGF treatments in our tertiary referral hospital. IVI anti-VEGF therapy was also associated with better HbA1c control. It is suggested that DME patients can receive intravitreal VEGF inhibitors without inducing more renal impairment.

## 1. Introduction

Diabetes mellitus (DM) is a prevalent disease with significant comorbidities that not only affect the eyes but also cause cardiovascular disease, nephropathy, and neuropathy. Diabetic retinopathy (DR) affects an estimated one in three people with DM and may result in severe visual impairment [1]. Diabetic macular edema (DME), a common complication of DR, is pathologically linked to the disruption of the blood–retinal barrier [2]. In the hypoxic microenvironment of DR, vascular endothelial growth factor (VEGF) leads to the formation of new blood vessels in the retina that have increased capillary permeability [3].

Currently, intravitreal injections (IVI) of VEGF inhibitors, including ranibizumab (Lucentis, Genentech Inc., South San Francisco, CA, USA) and aflibercept (EYLEA-Regeneron Pharmaceuticals, Inc., Tarrytown, New York, NY, USA and Bayer Healthcare Pharmaceuticals, Berlin, Germany) are mainstream and guideline-indicated treatments for diabetic macular edema (DME) [4]. However, recently published data have suggested that IVI of anti-VEGF may result in systemic absorption and leads to a further reduction in plasma VEGF activity, which in turn leads to accelerated hypertension, worsening proteinuria, glomerular disease, thrombotic microangiopathy, and possible chronic renal function decline [5,6].

There have been no previously published studies that discuss the impact on HbA1c of VEGF inhibitors. Kakizawa et al. described that poor glycemic control is correlated with increased levels of plasma VEGF, which may result in hypertension and vascular complications in diabetes [7]. However, Hanna et al. reported that IVI of anti-VEGF may result in systemic absorption and leads to a reduction in plasma VEGF activity [5]. We assume that systemic absorption by IVI of VEGF inhibitors may alter glycemic control and HbA1c levels.

Our study harnessed the China Medical University Hospital-Clinical Research Data Repository (CMUH-CRDR) to analyze real-world data on glycemic control (glycated hemoglobin A1c, HbA1c) and renal function (estimated glomerular filtration rate, eGFR) to assess their changes between patients receiving IVI treatment (including Ranibizumab and Aflibercept) and patients without IVI treatment.

## 2. Methods

### 2.1. Data Source

This study was based on the CMUH-CRDR from China Medical University Hospital (CMUH). At the time of this study, it contained medical records of 2,918,323 patients who were treated at CMUH between 2013 and 2019. Disease diagnoses, medical records, laboratory measurements, and physiological tests in the CMUH-CRDR were verified and validated [8]. This study was approved by the Big Data Center in CMUH and the Institutional Review Board of China Medical University Hospital (CMUH110-REC1-050(AR-2))—23 December 2021.

### 2.2. Study Subjects

We collected data from 18,251 patients with diabetic macular edema (DME) who had documented visits to CMUH between 2013 and 2019. We excluded 1086 patients without ophthalmology visits. DME patients were split into two groups based on ranibizumab and aflibercept treatment. DME patients with a history of ESRD and bevacizumab treatment, or those without recorded glycated hemoglobin A1c (HbA1c) levels or estimated glomerular filtration rate (eGFR), were excluded. The date of initial treatment was defined as the index date. The details are presented in Figure 1. A treatment patient was matched with a non-treatment DME patient by age (±2 years), gender, and the year of the clinic visit.

### 2.3. Measurement

The outcomes of interest were HbA1c levels and eGFR at baseline, 3 months, and 12 months after the index date. eGFR was estimated using the abbreviated Chronic Kidney Disease Epidemiology Collaboration (CKD-EPI) equation [8].

### 2.4. Statistical Analysis

For demographic data, age was expressed in range (20–64 and 65+) with the format of mean and standard deviation (SD). The data collection period was expressed with ‘year’ as the unit. Chi-squared tests and *t*-tests were used to test the differences in age, gender, and index-year between treatment and non-treatment patients. Due to subject matching, to test the difference between HbA1c and eGFR, paired sample t-tests between two groups at baseline, 3 months, and 12 months after the index date were utilized. Because the generalized estimating equation (GEE) model is a useful method for analyzing longitudinal data, in this study, we use GEE to analyze possible different serial changes (three different time points in each group) for HbA1c and eGFR with the number of injections of anti-VEGF. The SAS software version 9.4 (SAS Institute, Cary, NC, USA) was used to analyze, and the two-tailed test *p* < 0.05 was considered to be statistically significant.

## 3. Results

After matching, all 154 patients who received treatment and 154 patients who did not receive treatment were collected. There were no significant differences in age, gender, and treatment-year between the two groups. In those with treatment, the mean age was 61.8 years old (standard deviation = 10.6), and there was a slightly greater proportion of men (55.2% vs. 44.8%) (Table 1). In treatment patients, there were 24.0% of patients with one anti-VEGF treatment, 45.5% with two, and 29.5% with three or more.

### 3.1. HbA1c

Mean HbA1c decreased between baseline (7.72 ± 1.49% and 8.05 ± 1.92%) and 12 months after index date (7.28 ± 1.20% and 7.74 ± 1.55%) in both the treatment and non-treatment groups, respectively (trend *p* < 0.05) (Table 2). Compared to the non-treatment group, the treatment group had significantly lower HbA1c at 3 and 12 months after the index date. In the GEE model, HbA1cdecreased by 0.16% per anti-VEGF treatment (−0.16% and −0.16%, 95% CI = −0.29 to −0.03 and −0.30 to −0.03 in the crude and adjusted model) (Table 3). HbA1c decreased 0.19% over time (95% CI = −0.27, −0.10) in the adjusted GEE model.

### 3.2. eGFR

Mean eGFR decreased between baseline (66.1 ± 38.7 and 64.2 ± 38.4 mL/min/1.73 m^2^) and 12 months after index date (55.9 ± 34.8 and 60.2 ± 36.0 mL/min/1.73 m^2^) in both the treatment and non-treatment groups, respectively (trend *p* < 0.05) (Table 2). At each of the three time-points, there was no significant difference in eGFR between the two groups. After adjusting for age, gender, and index-year, eGFR in patients receiving treatment was still comparable to those without treatment (Table 3). For eGFR, there was no significant association with the number of anti-VEGF treatments. eGFR significantly decreased with age (−0.77 mL/min/1.73 m^2^, 95% CI = −1.22, −0.32) and time (−3.54 mL/min/1.73 m^2^, 95% CI = −4.66, −2.42) in the adjusted GEE model.

## 4. Discussion

In our study, we found that HbA1c levels in the treatment group were significantly lower than in the non-treatment group. Previous studies have described that VEGF levels in plasma are positively correlated with HbA1c levels [9,10]. Hanefeld et al. also reported that increased serum and plasma levels of VEGF in T2DM significantly depend on how well-controlled HbA1c levels are [11]. Moreover, Hanna et al. mentioned that IVI of VEGF inhibitors can lead to significant systemic absorption and measurable reduction in plasma VEGF activity [5]. Thus, we presume in our result that plasma VEGF blockage by the systemic absorption of IVI of anti-VEGF may improve glycemic control and HbA1c levels.

Previous studies have suggested that systemic anti-VEGF therapy is associated with renal function impairment [12,13]. However, other studies have disputed the relationship between nephrotoxicity and intravitreal VEGF inhibitors [5,6,14,15,16]. There are a number of population studies showing that intravitreal anti-VEGF agents are associated with nephrotoxicity and with increased mortality [5]. Hanna et al. described three cases of eGFR decline after intravitreal VEGF inhibitor [14]. Nobakht et al. reported one case of kidney function gradually declining after 148 administrations of intravitreal ranibizumab, bevacizumab, and aflibercept injections, eventually resulting in a need for hemodialysis [15]. Kakeshita et al. reported one case of renal focal segmental glomerulosclerosis after intravitreal aflibercept [16].

However, Glassman et al. reported no differences in changes in blood pressure or urine albumin-creatinine ratio as a reflection of kidney function in patients with DME treated with aflibercept, bevacizumab, or ranibizumab [17]. Kameda et al. showed that mean eGFR did not change after intravitreal administration of any of the three VEGF inhibitors [18]. Our study also demonstrated that mean eGFR did not change after intravitreal ranibizumab or aflibercept. These results suggest that DME patients can receive intravitreal VEGF inhibitors without inducing more renal impairment.

The administration of an intravitreal anti-VEGF medication results in small but measurable systemic levels of the drug [19,20]. The original FDA data also reported detectable serum levels around 0.2 nmol/L for aflibercept and 0.05 nmol/L for ranibizumab after intravitreal injection [5]. The reduction in plasma free-VEGF levels is associated with elevated levels of circulating anti-VEGF agents [19]. Our study showed that eGFR decreased with age and time and was not associated with anti-VEGF treatment significantly in these DME patients. However, a previous study showed that IVI bevacizumab, ranibizumab and aflibercept can cause the systemic suppression of VEGF, which might induce systemic adverse effects, including cardiovascular and arterial thromboembolic effects, renal and gastrointestinal effects, and wound-healing complications [19]. Hanna RM et al. also reported that thrombotic microangiopathy-associated nephrotoxicity could be induced by intravitreal VEGF inhibitors [5,6,21]. Therefore, it is reasonable to compare the changes in renal function in patients who receive IVI VEGF inhibitors or not. According to the recommendation of Hanna et al., if patients have increased creatinine and BUN by more than 25%, increased blood pressure by more than 20 mmHg, and increased urine protein-to-creatinine ratio by more than 25%, the dosage and frequency of intravitreal anti-VEGF therapy should be reduced [5]. If renal function still declines after a reduction in treatment or glomerular pathology demonstrates thrombotic microangiopathy, IVI anti-VEGF therapy should be suspended [5].

Our study has some limitations. First, most of the limitations of this study came from its retrospective nature. A more large-scale prospective design is needed to confirm our findings. Second, due to the inclusion and exclusion criteria, there were a relatively low number of study subjects. We have collected all available and eligible cases. Therefore, we did not set up a minimally required sample size because the main purpose of this study is not to compare treatment outcomes between exposure and non-exposure groups, for which sample size is crucial in the study design to make sure the power of the study is enough to confirm the value of the intervention. Furthermore, we could not separate treatment-naïve patients from non-treatment-naïve patients. Third, we followed eGFR and HbA1c for only one year. We may require a longer follow-up period to reach further conclusions. Fourth, the treatment group with better HbA1c may be biased by the reimbursement regulation of the Taiwanese National Health Insurance system. DME treatment with IVI of ranibizumab and aflibercept was only covered by the Taiwanese National Health Insurance system when HbA1c was less than 10%. This could cause bias in our result that HbA1c levels in the treatment group were significantly lower than in the non-treatment group.

## 5. Conclusions

This study shows that among DME patients in our tertiary care hospital who received IVI anti-VEGF, eGFR decreased with age and time and is not related to IVI anti-VEGF. IVI anti-VEGF therapy is also associated with better HbA1c control. It is suggested that diabetic patients can receive IVI VEGF inhibitors safely without significant renal function decline. Nevertheless, since previous studies reported adverse effects on renal function after therapy of IVI VEGF inhibitors [5,6,14,15,16], it is reasonable to monitor renal function in patients receiving IVI anti-VEGF therapy regularly. Further prospective studies are required to confirm our results and elucidate the systemic effects of IVI anti-VEGF therapy.

## Figures and Tables

**Figure 1 medicina-58-01081-f001:**
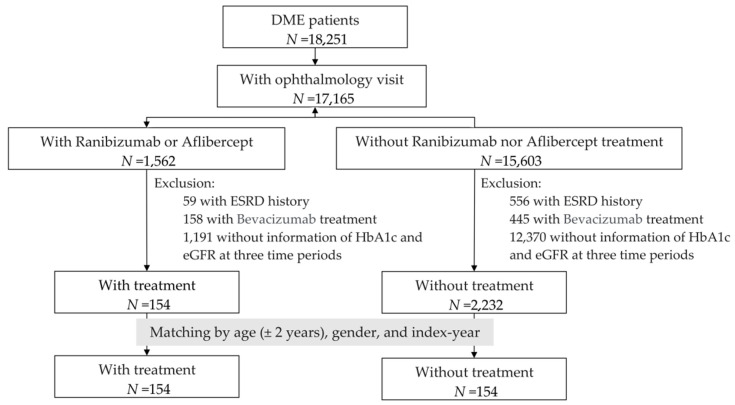
Flow chart for study subjects. Abbreviations: DME, diabetic macular edema; ESRD, end stage renal disease; HbA1c, glycated hemoglobin A1c; eGFR, estimated glomerular filtration rate.

**Table 1 medicina-58-01081-t001:** Distribution of demographics in study subjects.

Variable	Treatment Group *N* = 154		Non-Treatment Group *N* = 154		*p*-Value
	*n*	%	*n*	%	
Age, year					0.817
20–64	89	57.8	91	59.1	
65+	65	42.2	63	40.9	
Mean (SD)	61.8	(10.6)	61.9	(10.5)	0.953
Sex					1.000
Women	69	44.8	69	44.8	
Men	85	55.2	85	55.2	
Year					1.000
Number of injections					
1	34	24.0			
2	70	45.5			
3	43	27.9			
4	4	2.60			

Chi-square test, and *t*-test. SD, standard deviation.

**Table 2 medicina-58-01081-t002:** Mean of HbA1c and eGFR among time period in both groups.

Outcome	Treatment Group *N* = 154		Non-Treatment Group *N* = 154		*p*-Value
	Mean	SD	Mean	SD	
HbA1c, %					
Baseline	7.72	1.49	8.05	1.92	0.089
3 months	7.39	1.44	7.92	1.76	0.002
12 months	7.28	1.20	7.74	1.55	0.005
*p*-value	0.0004		0.0088		
eGFR, mL/min/1.73 m^2^					
Baseline	66.1	38.7	64.2	38.4	0.658
3 months	62.2	35.2	61.7	32.2	0.903
12 months	55.9	34.8	60.2	36.0	0.300
*p*-value	<0.0001		0.0004		

HbA1c, glycated hemoglobin A1C; eGFR, estimated glomerular filtration rate; SD, standard deviation.

**Table 3 medicina-58-01081-t003:** Result of a generalized estimating equation (GEE) model testing outcome with the number of injections.

Outcome	HbA1C, %			
	Crude (95% CI)	*p*-Value	Adjusted (95% CI)	*p*-Value
Number of injections	−0.16 (−0.29, −0.03)	0.015	−0.16 (−0.30, −0.03)	0.015
Age, year	−0.01 (−0.02, 0.01)	0.315	0.00 (−0.02, 0.01)	0.278
Men vs. women	−0.05 (−0.35, 0.26)	0.758	−0.05 (−0.35, 0.24)	0.725
Time period	−0.19 (−0.27, −0.10)	<0.0001	−0.19 (−0.27, −0.10)	<0.0001
	eGFR, mL/min/1.73 m^2^			
Number of injections	−0.42 (−3.64, 2.80)	0.797	−0.50 (−3.84, 2.63)	0.753
Age, year	−0.76 (−1.20, −0.32)	0.0007	−0.77 (−1.22, −0.32)	0.0009
Men vs. women	0.56 (−7.30, 8.42)	0.889	−1.08 (−8.90, 6.85)	0.790
Time period	−3.54 (−2.42, −6.20)	<0.0001	−3.54 (−4.66, −2.42)	<0.0001

## Data Availability

All data generated or analysed during this study are included in this published article.

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
