# Peer review of "Changes of Estimated Glomerular Filtration Rate and Glycated Hemoglobin A1c in Diabetic Macular Edema Patients Treated by Ranibizumab and Aflibercept in the Tertiary Referral Hospital"

_medicina, 2022, doi:10.3390/medicina58081081_

Round 1

Reviewer 1 Report

The number of injections of anti-VEGF drugs should be clear. Was there a correlation between the number of injections and HbA1c or GFR?

Taiwan's national health insurance system only covered patients with HbA1c less than 10%, which is a major economic bias in this study.

The lower HbA1c in the treatment group at 3 and 12 months may be an effect of anti-VEGF treatment.

Although this study is based on big data, the fact that HbA1c and GFR were lower in the treated and non-treated groups, respectively, does not imply a small effect of anti-VEGF drugs.

Reviewer 2 Report

Introduction:

- The author evaluated HBA1C but there is no information regarding the relation between HBA1C and VEGF? and why do the authors need to evaluate HBA1C in DME patients treated with VEGF?

- the intro is too short

Results:

- The author mentioned "There were no significant differences in age, gender, and 104 treatment-year between two groups", but you already matched at the beginning, so it is not surprising that no statistical significance was observed. and this was similar to gender. although the proprotion was higher but did not statistically significant. 

- In Table 1, missing p-value, only two groups (2013-2015 and 2016-2019) how about the previous years, because in the method you mentioned that the data was collected from 2003-2019??

Discussion:

- Again, missing information about HBA1C, and no discussion related to HBA1C.

- the author mentioned "a previous study showed that IVI of ranibizumab and aflibercept can bypass the blood-retinal barrier, resulting in the detection of these anti-VEGF medications in the glomeruli of monkeys.13 Therefore, it is reasonable to monitor the renal function of patients 160 who receive IVI of VEGF inhibitors." Based on the animal study? I am not sure whether it is sufficient to support your statement about monitoring renal function. 

- you mentioned "Under the effect of patients’ self-awareness of DME and active referral by the physicians, we could explain why a greater proportion of treatments occurred from 2016-2019." what is the relation with the finding of the study.?

- the author mentioned "Second, due to the strict inclusion and exclusion criteria", I am not sure whether strict is the correct term because most of the record doesn't have complete data?

Round 2

Reviewer 1 Report

The authors complied with the reviewers' requests and the paper is now better.

Author Response

Thank you so much!

Reviewer 2 Report

Is there any explanation why locally injected VEGF caused systemic manifestation? I think it is good to elaborate that way, despite negative findings on GFR. 

Author Response

Yes, thank you for your input. We have added the explanation in our manuscript.

The administration of an intravitreal anti-VEGF medication results in small but measurable systemic levels of the drug. The original FDA data also reported detectable serum levels around 0.2 nmol/l for aflibercept and 0.05 nmol/l for ranibizumab after intravitreal injection. The reduction in plasma free-VEGF levels are associated with elevated levels of circulating anti-VEGF agents. Our study showed that eGFR decreased with age and time, and was not associated with anti-VEGF treatment significantly in these DME patients. However, a previous study showed that IVI bevacizumab, ranibizumab and aflibercept can cause the systemic suppression of VEGF, which might induce the systemic adverse effects, include cardiovascular and arterial thromboembolic effects, renal and gastrointestinal effects, and wound-healing complications. Hanna RM et al. also reported that thrombotic microangiopathy-associated nephrotoxicity could be induced by intravitreal VEGF inhibitors. Therefore, it is reasonable to compare the changes of renal function in patients who receive IVI VEGF inhibitors or not.